# Isolation of *Latilactobacillus curvatus* with Enhanced Nitric Oxide Synthesis from Korean Traditional Fermented Food and Investigation of Its Probiotic Properties

**DOI:** 10.3390/microorganisms11092285

**Published:** 2023-09-11

**Authors:** Hao Jin, Sang-Kyu Park, Yong-Gun Yun, Nho-Eul Song, Sang-Ho Baik

**Affiliations:** Department of Food Science and Human Nutrition, Jeonbuk National University, Jeonju 54896, Republic of Korea; blueberry-73jinhao@naver.com (H.J.); dave0340@naver.com (S.-K.P.); shooyygboa@nate.com (Y.-G.Y.); nesong@kfri.re.kr (N.-E.S.)

**Keywords:** Jangajji, lactic acid bacteria, probiotic, *Latilactobacillus curvatus*, nitric oxide

## Abstract

Nitric oxide (NO) is a free radical associated with physiological functions such as blood pressure regulation, cardiovascular health, mitochondrial production, calcium transport, oxidative stress, and skeletal muscle repair. This study aimed to isolate *Latilactobacillus curvatus* strains with enhanced NO production from the traditional Korean fermented food, jangajji, and evaluate their probiotic properties for industrial purposes. When cells were co-cultured with various bacterial stimulants, NO production generally increased, and NO synthesis was observed in the range of 20–40 mg/mL. The selected strains of *Lat. curvatus* were resistant to acid and bile conditions and with variable effectiveness (1–14%) in adhering to Caco-2 cells. Most bacterial strains can inhibit the growth of various pathogens. In addition, they are capable of reducing cholesterol levels via assimilation of cholesterol at 10–50%. Among the selected NO synthases from *Lat. curvatus* strains, the strain JBCC38 showed the highest capacity to scavenge ABTS (30.1%) and DPPH radicals (39.4%). Moreover, these strains exhibited immunomodulatory properties. The production of TNF-α and IL-6 in the macrophages treated with various bacterial stimulants was induced in all the selected strains.

## 1. Introduction

Nitric oxide (NO) is a small signaling molecule that has various functions in immune responses [1]. NO is an important component of the immune system because it is involved in pathogen defense [2], inflammation regulation [3], vasodilation [4], and antiviral activity [5]. Its versatile functions make it essential for maintaining immune homeostasis and mounting effective immune responses. Some strains of lactic acid bacteria (LAB) can produce NO via nitric oxide synthase [6] enzymes and are of interest because of their potential health benefits, such as antimicrobial and anti-inflammatory properties [7]. The NO produced by LAB modulates immune responses and exerts immunomodulatory effects, according to previous research [8]. Overall, the relationship between NO and LAB is complex and context-dependent. Because NO production by LAB as probiotics can affect their metabolism, antimicrobial activity, and immunomodulatory effects, further research is needed to fully understand the impact of the interaction between NO and LAB for its potential applications in various fields, including medicine, food science, and agriculture.

*Lat. curvatus* is a candidate probiotic that has attracted considerable attention because of its excellent fermentation properties and health benefits. *Lat. curvatus* is commonly found in various environments, including fermented foods, such as meat, vegetables, and dairy products [9]. In essence, *Lat. curvatus* is useful in the food processing industry because it produces lactic acid for the preservation and flavor development of fermented dairy foods. In addition, it has been on focused on due to the production of class II bacteriocins, such as curvacin A and Sakacin P, by *Lat. curvatus* for their anti-pathogenic properties [10]. Moreover, *Lat. curvatus* produces organic acids that are used for reducing the pH of meat product fermentation systems, resulting in reduced nitrite content in meat products [11]. *Lat. curvatus* also exhibits high fatty acid-hydrolyzing activity that is used for the development of desirable flavors in meat products during fermentation and curing processes [12]. *Lat. curvatus* has a high potential to benefit human health in terms of obesity [13], anti-inflammation, anti-hyperlipidemia [14], and prevention of the muscle atrophy induced by dexamethasone [15].

In this study, we aimed to isolate *Lat. curvatus* with enhanced NO synthesis from a traditional Korean high-salt fermented pickle, jangajji, and evaluate its probiotic properties for potential application in promoting immune health. A traditional fermented food, jangajji is very popular as a side dish in Korea; it is made from various vegetables pickled by using high-salt brine and developing various lactic acid bacterial consortia. The findings of this study can provide valuable insights into the use of *Lat. curvatus* strains as probiotics with immunomodulatory activities, thereby contributing to our understanding of their potential health benefits.

## 2. Materials and Methods

### 2.1. Jangajji Samples and Lactic Acid Bacteria Isolation

Three jangajji samples containing the basic materials of chili pepper jangajji, perilla leaf jangajji, and garlic jangajji were purchased from the local markets in the Jeonju region (Chonbuk, Republic of Korea). The samples were homogenized at 500 rpm and suspended in sterile phosphate buffer solution (PBS, 50 mM, pH 7.0). Subsequently, the diluted solution was aseptically distributed for 24 h at 37 °C on lactobacilli MRS agar (Difco Laboratories, Detroit, MI, USA) plates with 1% (*w*/*v*) CaCO_3_. Colonies were selected at random and cultivated in MRS broth at 37 °C for 18 h. *Lacticaseibacillus rhamnosus* GG (ATCC 53103, LGG), a representative probiotic strain, was used as the reference control to evaluate the probiotic properties of the isolated strains. For further analysis, the isolates were stored at −80 °C in MRS broth containing 20% (*v*/*v*) sterile glycerol.

### 2.2. Identification of the Isolated Strains

Sequencing analysis of 16S rDNA was performed as previously described [16]. By comparing the obtained sequences to the LAB sequences in the DNA Databases (http://www.ncbi.nih.gov/BLAST, accessed on 13 April 2022), homology analysis was performed.

### 2.3. Probiotic Properties

#### 2.3.1. Cell Viability and Nitrite Oxide Analysis

Cytotoxicity was performed using the methylthiazolyltetrazolium (MTT) assay method described by Levy and Simon [17]. RAW 264.7 cells (seeded at 5 × 10^4^ CFU/mL) were seeded into 96-well plates (Nunc, Roskilde, Denmark) and incubated for 24 h. After treatment for 1 h with or without lipopolysaccharides (LPS: 1 g/mL), the MTT assay was performed, and the results were expressed as a percentage of the respective controls.

NO was produced by identical RAW264.7 cells at a concentration of 5 × 10^4^ CFU/mL. They were seeded in 96-well culture plates and incubated for 2 h. To each well, either viable or heat-inactivated bacteria (100 μL) were added. After 24 h incubation (37 °C, 5% CO_2_), the supernatant was collected and analyzed for NO via the Griess reaction [18]. The NO concentrations were calculated based on a standard curve prepared using sodium nitrite.

#### 2.3.2. Determination of Cytokines (TNF-α and IL-6)

The concentrations of interleukin-6 (IL-6) and tumor necrosis factor-α (TNF-α) were assessed using commercial ELISA kits (Pharmingen, CA, USA) according to the manufacturer’s recommendations, and absorbance was measured at 450 nm using a 96-well plate reader.

#### 2.3.3. Acid Resistance and Bile Tolerance

The acid resistance of selected LAB was determined using the 2005 method [19]. The survival rate of LAB strains was calculated as a percentage, indicating the extent of survival after exposure to acid or bile. The survival rate was calculated as follows:Survival rate (%): (log A1/A0) × 100%(1)

A1: Total viable count of probiotic strains after treatment; A0: total viable count of probiotic strains before treatment.

#### 2.3.4. Bile Salt Hydrolase (BSH) Activity

The presence of BSH activity was typically indicated by the appearance of a clear zone around the LAB colonies on the BSH screening medium, indicating hydrolysis of the sodium salt, taurodeoxycholic acid (TDCA, Sigma Aldrich, St Louis, MO, USA). The BSH screening medium was prepared by adding TDCA and CaCl_2_ to the MRS agar. The concentration of TDCA was 0.5% (*w*/*v*), and CaCl_2_ was added at a concentration of 0.37 g/L [20]. TDCA is a bile salt that serves as a substrate for BSH activity, whereas CaCl_2_ provides essential calcium ions. Sterile discs were put on the BSH screening medium, and 100 µL of the selected LAB strains culture were dropped onto them.

#### 2.3.5. Cholesterol Assimilation

The ability of *Lactobacillus* to assimilate cholesterol was measured according to a previously described method, with some modifications [21,22]. Cholesterol assimilation was calculated using the following equation: Cholesterol assimilation (%) = (1 − A1/A0) × 100(2)

A1: Cholesterol represents MRS at 0 h; A0: cholesterol represents MRS at 24 h.

### 2.4. Probiotic Adhesion to Caco-2 Cells

In vitro adhesion assays were performed according to the methods described in previous studies with some modifications [23]. An in vitro adhesion assay was performed as described by Kashiwagi et al. [24]. We initially seeded Caco-2 cells into each well at 5 × 10^4^ cells/mL. The number of Caco-2 cells in each well was determined using a hematometer. The adhesion to Caco-2 cells was calculated using the following equation:Adhesion to Caco-2 cells (%) = 100 × (*Abs Initial* − *Abs final*)/*Abs Initial*(3)
= Δ*Abs*/*Abs Initial* × 100(4)

### 2.5. Cell Surface Hydrophobicity

To determine cell surface hydrophobicity, bacterial adhesion to hydrocarbons was measured as previously described [25]. The microbial adhesion to hydrocarbon (MATH) test is sensitive to a variety of variables, including culture period, cultivation medium composition, presence of certain acids, and solvent type [26]. This method involves assessing the affinity of microorganisms for nonpolar solvents, such as hexane, xylene, or toluene. Hydrocarbons were extracted in this study. Surface hydrophobicity was calculated as the percentage decrease in the absorbance of the aqueous phase after mixing and phase separation, relative to that of the original suspension (AbsInitial), using the following equation:Surface Hydrophobicity (%) = 100 × (*Abs Initial* − *Abs final*)/*Abs Initial*
Δ*Abs*/*Abs Initial* × 100(5)

### 2.6. Cell Aggregation

The auto-aggregation assay was performed according to the method described by Farid et al. [27]. The percentage difference between the initial and final absorbance values provides an index of cellular auto-aggregation, which can be expressed by the following equation:Aggregation (%) = 100 × (*Abs Initial* − *Abs final*)/*Abs Initial*(6)
= Δ*Abs*/*Abs Initial* × 100(7)

### 2.7. Antibiotic Resistance

To test antibiotic resistance, bacterial strains were inoculated at a concentration of 1% (*v*/*v*) in MRS broth supplemented with various final concentrations of antibiotics. Ampicillin and vancomycin were used as inhibitors of cell wall synthesis, and kanamycin and chloramphenicol were used as inhibitors of protein synthesis (Sigma Chemical Co., St. Louis, MO, USA). The concentrations tested were 2, 4, 8, 16, 32, 64, 128, 256, 512, and 1024 mg/mL and were examined in triplicate for growth in a microplate reader (OD at 580 nm) following a 24 h incubation period at 37 °C. The minimal inhibitory concentration (MIC) was determined using MRS broth and was defined as the lowest concentration of antibiotics that completely inhibited visible growth compared to an antibiotic-free control well.

### 2.8. Antimicrobial Activity

The antimicrobial activity was determined using a paper disc according to the method described by Wu et al. [28], with minor modifications. A total of 50 µL of the supernatants of LAB strain were dropped onto the discs on nutrient broth agar plates. Each indicator pathogenic strain was grown in nutrient broth at 37 °C for 24 h, and sterile 8-mm paper discs with LAB were placed on the agar. After the plates were incubated for 24 h at 37 °C, the diameter of the clear zone was measured.

### 2.9. Antioxidant Activity

From the measurement of 2,2′-azinobis (3-ethylbenzothiazoline-6-sulfonic acid) (ABTS) radical scavenging activity, 2,2-diphenyl-1-picrylhydrazyl (DPPH) radical scavenging activity, and superoxide dismutase (SOD) activity, the antioxidant activity was determined. The ABTS free radical scavenging activity was measured at 734 nm after the reaction [29]. The DPPH radical scavenging activity of the LAB was determined according to the method described by Li et al. (2013). The SOD activity of LAB was determined via spectrophotometry according to the method described in a previous study, with some modifications [30].

### 2.10. Statistical Analysis

All experiments were performed in triplicate. The error bars in the graphs indicate standard deviation. Data were analyzed using analysis of variance [31] and Student’s *t*-test. Differences were considered significant at *p* < 0.05.

## 3. Results and Discussion

### 3.1. Isolation of Latilatobacillus curvatus from Jangajji and Their Identification

A total of 110 putative LAB strains were isolated from three types of jangajji samples: garlic jangajji (one strain), hot pepper jangajji (14 strains), and perilla leaf jangajji (95 strains). Amplified ribosomal DNA restriction analysis was performed to verify the LAB strains. The six strains categorized as *Lat. curvatus* (JBCC38, JBCC47, JBCC60, JBCC65, JBCC93, and JBCC99) were selected(Data not shown here). When we examined the similarity of the 16S rDNA sequences, all the selected strains showed high sequence similarity TO *Lat. curvatus* (99–100%), as shown in Table 1.

### 3.2. Evaluation of Nitric Oxide Synthesis Activity of Lat. curvatus JBCC Strains

The MTT assay is commonly used to assess cell viability and cytotoxicity in various biological and biomedical research applications [32]. When we assessed the effect of selected *Lat. curvatus* strains on cell viability, no differences were observed, indicating that the selected strains had no cytotoxic effect on macrophages (Figure 1a). To evaluate the NOS activity of the *Lat. curvatus* JBCC strain, RAW 264.7 cells and heat-killed *Lat. curvatus* JBCC strains were used. The results indicated that the six heat-killed *Lat.* curvatus JBCC strains significantly induced NO synthesis. In particular, *Lat. curvatus* JBCC93, JBCC65, and JBCC47 increased NO synthesis approximately 10-, 9-, and 6-fold, respectively, compared with the LPS-treated positive control (*p* < 0.05), as shown in Figure 1b. Among the examined *Lat. curvatus* strains, the most highly active ones were *Lat. curvatus* JBCC93 (40 μg/mL) and *Lat. curvatus* JBCC65 (35 μg/mL), which exhibited almost a 2-fold higher synthesis rate than that of the commercial probiotic strain, *Lcb. rhamnosus* GG (27 μg/mL). However, only *Lat. curvatus* JBCC47 showed significantly higher NO production (25 μg/mL) than the LPS-treated positive control (20 μg/mL). To the best of our knowledge, the activity of isolated *Lat. curvatus* strains in this study may have been higher than that previously documented. It was reported that NO production in RAW264.7 cells by *Lat. curvatus* isolated in kimchi was 22.35–24.11 μM, which is a very low level compared to that produced by our selected *Lat. curvatus* strains [33].

### 3.3. Probiotic Properties: Tolerance of Acid and Bile Salt in the Selected Latilactobacillus curvatus Strains

As shown in Table 1, all six *Lat. curvatus* strains showed a relatively high survival rate (84–95%) at pH 3, which is similar to that of *Lcb. rhamnosus* GG. All the selected *Lat. curvatus* strains, as well as *Lcb. rhamnosus GG*, showed a critically decreased survival rate with decreasing pH. *Lat. curvatus* JBCC65 (83.7%), *Lat. curvatus* JBCC38, and JBCC60 were also tolerant to 3% (*w*/*v*) acid even after 24-h exposure as shown in Table 1. None of the *Lat. curvatus* strains exhibited high BSH activity. Although *Lat. curvatus* JBCC38 and JBCC60 had relatively high BSH activities, they were lower than those of *Lcb. rhamnosus* GG. BSH is an enzyme that can hydrolyze bile salts, reducing their detergent-like properties and enhancing bacterial survival in the presence of bile. However, there seems to be little relationship between BSH activity and tolerance.

To determine the probiotic activity of *Lat. curvatus* strains, cholesterol-lowering activity was examined because the BSH activity in probiotic strains has been associated with a cholesterol-lowering potential [34] and is a crucial indicator for the selection of probiotic strain adjuncts to manage hypercholesterolemia. As shown in Figure 2, except for *Lat. curvatus* JBCC65 and JBCC93 (50% and 56.2%, respectively), the examined *Lat. curvatus* strains exhibited cholesterol-lowering effects. JBCC38, JBCC47, and JBCC60 showed a cholesterol removal rate higher than 65%, similar to that of *Lcb. rhamnosus* GG, indicating that they have a good potential to lower cholesterol levels. *Lat. curvatus* JBCC99 removed cholesterol up to 62.9%, but this rate was lower than that of control strain of *Lcb. rhamnosus* GG. Conclusively, *Lat. curvatus* strains isolated in this study are good probiotic candidates.

### 3.4. Adhesion, Surface Hydrophobicity, and Auto-Aggregation

As shown in Figure 3a, all tested strains adhered from 10% to 14% under in vitro conditions, except *Lat. curvatus* JBCC38 and JBCC99. This was a higher rate than that of the other LAB, such as *L. fermentum*, a strain isolated from fermented cereal food, 8.5% [35]. When we examined the cell surface hydrophobicity, an important factor in cell adhesion [36], *Lat. curvatus* JBCC47 (68.6%) and JBCC38 (61.0%) are shown in Figure 3b. Other strains of *Lat. curvatus* also showed higher hydrophobicity than *Lcb. rhamnosus* GG (21%). The higher hydrophobicity of the cell surface might even lead to higher adhesion to Caco-2 cells, which is not always correlated with cell adhesion capacity. Although *Lat. curvatus* JBCC38 exhibited a very high hydrophobicity, its adhesion assay was only 1%. In contrast, *Lat. curvatus* JBCC60 showed a good adhesion capacity (35.3%) but low hydrophobicity (14%). Nonetheless, all *Lat. curvatus* strains had higher hydrophobicity than the control strain *Lcb. rhamnosus* GG (21%). This is in agreement with the hydrophobicity of other strains, such as *L. fermentum* and *L. casei*, which ranged from 0.3% to 68.8% [11]. Co-aggregation plays a role in the persistence of bacteria in the intestine and facilitates the adhesion process by enabling probiotic bacteria to form aggregates or clumps, which increases their ability to adhere to the gut lining [3]. All the strains tested in this study exhibited good co-aggregation properties, ranging from 24.2% to 37.8%, which were higher than those of the control strain, *Lcb. rhamnosus* GG (20.1%). *Lat. curvatus* JBCC38, JBCC93, and JBCC99 exhibited the highest auto-aggregation ability at 37.8%, 36%, and 31.2%, respectively (Figure 3c).

### 3.5. Antimicrobial Activity and Antibiotic Susceptibility

As shown in Table 2, the six selected strains exhibited diverse antimicrobial activities against different Gram-positive and Gram-negative pathogens of *Staphylococcus aureus*, *S. epidermidis*, and *S. xylosus* and Gram-negative strains of *Pseudomonas aeruginosa*, *P. putida*, *Propionibacterium acnes*, *Bacillus cereus*, *B. vallismortis*, and *Escherichia coli* [37,38]. An agar spot assay was used to study the antimicrobial activity of the six *Lat. curvatus* JBCCs, and all the strains inhibited the reference strains *S. aureus*, *S. epidermidis*, *S. xylosus*, and *B. cereus*. *Lat. curvatus* JBCC38 had no inhibitory effect on *Pro. acnes*. *Lat. curvatus* JBCC47 and JBCC99 did not inhibit *E. coli*. Compared to the other strains, *Lat. curvatus* JBCC99 showed a relatively weak inhibitory activity against the tested pathogens (Table 3). All the strains examined in this study were observed to be highly susceptible to ampicillin and chloramphenicol, as shown in Table 3, but were clinically resistant to vancomycin and kanamycin, even though there was a difference in the degree of resistance of each strain. In particular, *Lat. curvatus* JBCC38 was strongly resistant to vancomycin and kanamycin (MIC = 1024 μg/mL). *Lat. curvatus* JBCC47 and JBCC60 were resistance to kanamycin but susceptible to vancomycin (MIC = 512 μg/mL). *Lat. curvatus* JBCC93 was resistant to both vancomycin and kanamycin (MIC = 256 μg/mL). *Lat. curvatus* JBCC65 was susceptible to all the examined antibiotics but was clinically resistant to kanamycin (MIC = 32 μg/mL). Our results clearly show that the *Lat. curvatus* isolated in this study is resistant to various antibiotics and beneficial to human and animal health [39].

The bacterial strains used for the inhibition study were *Staphylococcus aureus* KCTC 1916, *Staphylococcus epidermidis* KCTC 1917, *Staphylococcus xylosus* KACC 13239, *Pseudomonas aeruginosa* KACC 10186, *Pseudomonas putida* KACC 10266, *Bacillus cereus* KACC 10097, *Bacillus vallismortis* KACC 12149, *Escherichia coli* KACC 12149, and *Propionicbacteria acnes* KCTC 3314.

### 3.6. Antioxidant Activity

As shown in Figure 4, *Lat. curvatus* JBCC38 showed a high capacity to scavenge ABTS (30.1%) and DPPH radicals (39.4%). *Lat. curvatus* JBCC47 showed the second highest capacity to scavenge DPPH (38.5%) and ABTS radicals (30.3%) compared to the standard probiotic strain *Lcb. rhamnosus* GG (30.8% and 39.2%, respectively). Compared to *Lcb. rhamnosus* GG, *Lat. curvatus* strains showed similar ABTS and DPPH radical scavenging activities, although no strains showed similar SOD activity (45.5%). The scavenging rates of ABTS radicals ranged from 25.7% to 30.9%, whereas those of DPPH radicals ranged from 17.9% to 39.1%. However, the SOD activities of these strains were relatively low, ranging from 13.6% to 19.9%. Because SOD targets superoxide radicals and unstable free radicals, such as ABTS or DPPH, its ability to scavenge or reduce these substrates is usually low [40]. While SOD might not be directly effective against stable free radicals such as ABTS or DPPH, its primary role is to neutralize superoxide radicals and contribute to maintaining overall cellular antioxidant balance by preventing excessive superoxide accumulation. Other antioxidant systems are responsible for dealing with a broader range of reactive oxygen species, including stable free radicals. Even though SOD may indirectly help the antioxidant defense system by eliminating superoxide radicals, it does not directly eliminate stable free radicals, such as ABTS or DPPH.

### 3.7. Induction of Cytokine Secretion

Cytokines play important roles in regulating inflammatory and immune responses. In particular, pro-inflammatory mediators, such as TNF-α and IL-6, are potent immunomodulatory cytokines of activated macrophages (Arango Duque and Descoteaux, 2014). The induction of these cytokines contributes to the maintenance of the state of controlled inflammation that occurs under normal conditions observed in the gut mucosa. Analysis of the released cytokines showed that LAB strains were different in how well they could trigger pro-inflammatory cytokines, such as TNF-α and IL-6. Figure 5 shows the pattern of cytokines induced by the strains isolated from jangajji on RAW264.7 cells and the concentrations of TNF-α and IL-6. The production of TNF-α in macrophages treated with various bacterial stimulants was significantly increased with all six selected strains compared to the untreated negative control (*p* < 0.05). Cells treated with the two strains of *Lat. curvatus* JBCC38 and JBCC60 showed significantly higher levels of TNF-α than did LPS-treated cells (*p* < 0.05). Four strains, *Lat. curvatus* JBCC38, JBCC60, JBCC65, and JBCC96, induced significantly higher levels of TNF-α compared to those induced by *Lcb. rhamnosus* GG. In addition, IL-6 levels were significantly higher in the cells treated with all six strains than in the untreated cells. Cells treated with the six isolates showed significantly lower IL-6 levels than LPS-treated cells did. However, cells treated with *Lat. curvatus* JBCC65, JBCC93, JBCC38, and JBCC60 showed a higher induction than those treated with *Lcb. rhamnosus* GG. Based on these results, three strains, *Lat. curvatus* JBCC38, JBCC60, and JBCC65, showed the most promising immunomodulatory activity.

Once activated, macrophages phagocytose microorganisms, release pro-inflammatory cytokines, and deliver antigens to helper T cells. The release of various substances mediates a substantial portion of these actions [41,42]. For instance, in the presence of pathogens, macrophages can produce NO to kill antigens, and activated macrophages secrete various cytokines, including IL-6 and TNF-α, thereby initiating and enhancing the immune response and other multi-faceted responses. Production of TNF-α in the macrophages stimulated by the selected strains was even greater than that in the cells treated with LPS, indicating that strains JBCC38 and JBCC47 could induce the secretion of higher levels of TNF-α by RAW 264.7 cells. IL-6 is involved in a wide range of immune processes, such as promoting the proliferation and differentiation of B cells, inducing acute-phase protein production, and coordinating the transition from innate to adaptive immunity. It also helps to regulate the balance between pro-inflammatory and anti-inflammatory responses [43]. In this study, we demonstrated the ability of selected strains to induce the production of IL-6 in RAW 264.7 cells. Our results showed that IL-6 production in the cells was significantly enhanced by some of the selected strains, such as *Lat. curvatus* JBCC38, JBCC60, JBCC65, and JBCC93, which showed higher levels than that of *Lcb. rhamnosus* GG. These observations suggest that IgA levels in the cells induced by the selected strains may increase, thereby facilitating the modulation of the host immune response. According to previous studies, intracellular metabolites or molecules from *L. plantarum* KFCC11389P inhibit the production of the pro-inflammatory cytokines IL-6 and TNF-α in LPS-stimulated RAW 264.7 macrophages. However, these anti-inflammatory effects were not detected when viable or heat-killed cells were used, suggesting that *L. plantarum* KFCC11389P cells did not affect anti-inflammatory cytokines and thus could not exert immune-modulating activities [44]. Future research is required to identify the active elements in the bacterial cell structure and confirm the anti-inflammatory activity in human clinical trials and animal disease models to gain a better understanding of the anti-inflammatory effects induced by particular *lactobacillus* strains.

## 4. Conclusions

During the course of our study, we screened *Lat. curvatus* JBCC from traditional Korean jangajji for augmented NO production and conducted an in vitro assessment of its probiotic potential. When cells were co-cultured with various bacterial stimulants, NO production generally increased. *Lat. curvatus* JBCC93 and JBCC65 showed a higher NO synthesis rate than the control strain, *Lcb. rhamnosus* GG. As a probiotic, the *Lat. curvatus* strains are tolerant to acid and bile conditions in an artificial gastrointestinal system. *Lat. curvatus* also showed a good adhesion to Caco-2 cells, pathogen inhibitory activity, cholesterol-lowering effects, and antioxidant activity. *Lat. curvatus* strains JBCC38, JBCC60, JBCC65, and JBCC93 showed substantially enhanced IL-6 production in RAW 264.7 cells compared to *Lcb. rhamnosus* GG. *Lat. curvatus* JBCC38 and JBCC47 induced secretion of higher levels of TNF-α by RAW 264.7 cells.

## Figures and Tables

**Figure 1 microorganisms-11-02285-f001:**
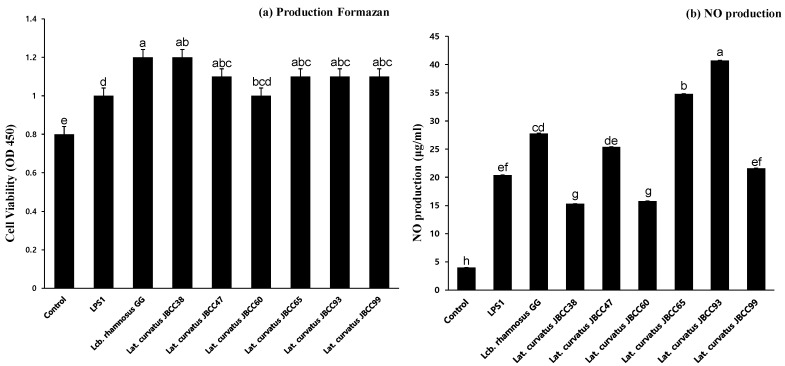
(**a**) Cell cytotoxicity of the selected strains; (**b**) NO production by RAW 264.7 cells treated with *Lat. curvatus* JBCC38, JBCC47, JBCC60, JBCC65, JBCCU93, JBCC99, and the reference strain, *Lcb. rhamnosus* GG. (Different lowercase letters above the bars indicate significant differences among groups; *p* < 0.05).

**Figure 2 microorganisms-11-02285-f002:**
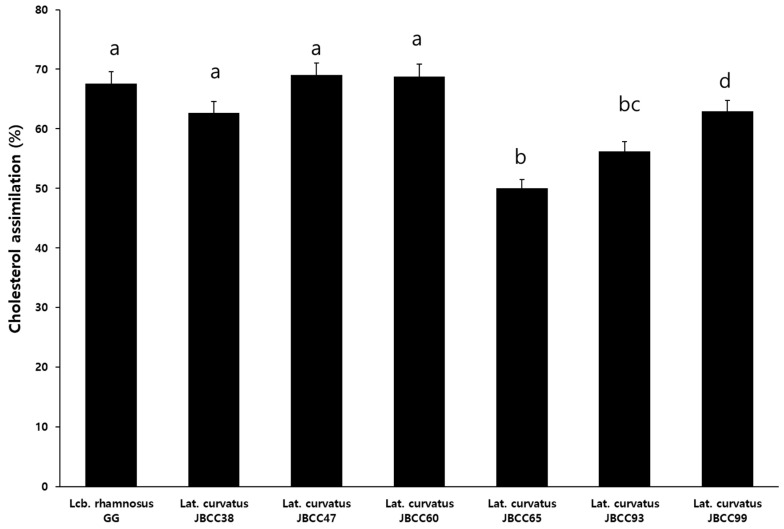
Cholesterol-lowering effects of *Lat. curvatus* JBCC38, JBCC47, JBCC60, JBCC65, JBCC93, JBCC99, and the reference strain, *Lcb. rhamnosus* GG. (Different lowercase letters above the bars indicate significant differences among groups, *p* < 0.05).

**Figure 3 microorganisms-11-02285-f003:**
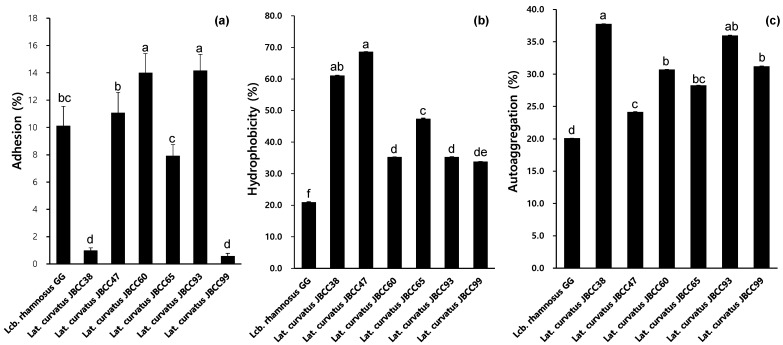
(**a**) Adhesion ability of the selected strains to Caco-2 cells; (**b**) Hydrophobicity (%); (**c**) Auto-aggregation (%) of *Lat. curvatus* JBCC38, JBCC47, JBCC60, JBCC65, and the reference strain, *Lcb. rhamnosus GG*. (Different lowercase letters above the bars indicate significant differences among groups; *p* < 0.05).

**Figure 4 microorganisms-11-02285-f004:**
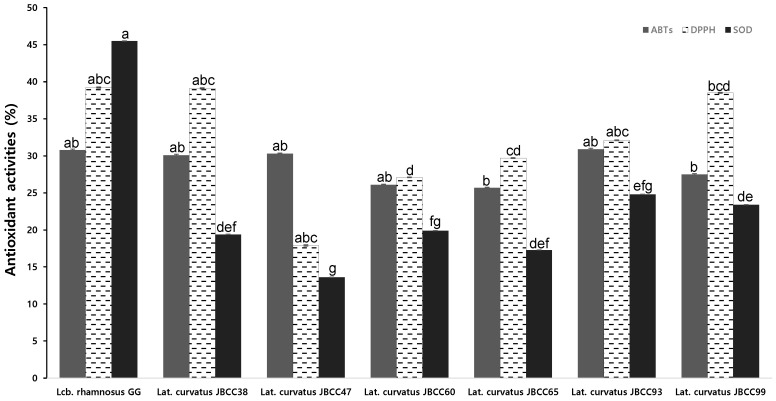
Evaluation of the antioxidant activities of *Lat. curvatus* JBCC38, JBCC47, JBCC60, JBCC65, JBCC93, JBCC99, and the reference strain, *Lcb. rhamnosus* GG. Different lowercase letters above the bars indicate significant differences among groups (*p* < 0.05).

**Figure 5 microorganisms-11-02285-f005:**
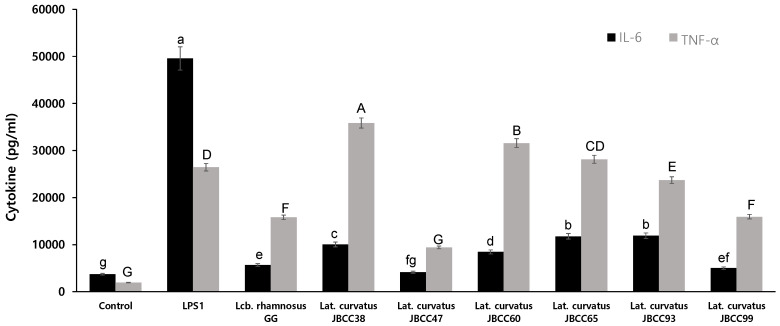
IL-6 and TNF-α production by RAW 264.7 cells with *Lat. curvatus* JBCC38, JBCC47, JBCC60, JBCC65, JBCC93, JBCC99, and the reference strain, *Lcb. rhamnosus* GG. (Different letters above the bars indicate significant differences among groups; *p* < 0.05.)

**Table 1 microorganisms-11-02285-t001:** Evaluation of the probiotic properties of the selected *Lat. curvatus* strains isolated from the traditional Korean fermented food, jangajji.

Identification	16S rDNA	Low pH (SR%) ^A^	Bile Salts (SR%) ^B^	BSH Activity ^C^	Cholesterol Assimilation (%)	Initial	pH Tolerance Mean Counts (log CFU/mL)					
						pH 7.0	pH 3.0	Survival Rate (%)	pH 2.0	Survival Rate (%)	pH 1.5	Survival Rate (%)
*Lcb. rhamnosus GG* (ATCC 53103) ^D^	100	95.4 ^c^	88.7 ^c^	+++++	67.6	9.04 ± 0.07	8.57 ± 0.12	94.80	2.71 ± 0.34	29.98	3.49 ± 0.08	38.61
*Lat. curvatus* JBCC38	99	84.1 ^f^	100.1 ^a^	+++	62.7	8.67 ± 0.12	7.28 ± 0.03	83.97	3.33 ± 0.04	38.41	3.69 ± 0.29	42.56
*Lat. curvatus* JBCC47	99	92.8 ^e^	92.7 ^b^	+	69.0	7.87 ± 0.19	7.26 ± 0.04	92.25	3.43 ± 0.03	43.58	2.82 ± 0.05	35.83
*Lat. curvatus* JBCC60	99	87.4 ^e^	98.7 ^a^	+++	68.8	8.41 ± 0.03	7.34 ± 0.01	87.28	2.50 ± 0.14	29.73	2.77 ± 0.02	32.94
*Lat. curvatus* JBCC65	99	95.5 ^c^	83.7 ^c^	+	50.0	8.48 ± 0.03	8.08 ± 0.02	95.28	3.50 ± 0.02	41.27	2.97 ± 0.04	35.02
*Lat. curvatus* JBCC93	99	88.9 ^e^	85.9 ^c^	+	56.2	8.45 ± 0.05	7.58 ± 0.06	89.70	2.79 ± 0.16	33.02	2.42 ± 0.06	28.64
*Lat. curvatus* JBCC99	99	94.0 ^d^	85.1 ^a^	+	62.9	8.34 ± 0.11	7.88 ± 0.04	94.48	2.44 ± 0.04	29.26	2.39 ± 0.11	28.66

^A^ Survival rate (SR) after 3 h in low pH. ^B^ Survival rate (SR) in 3% bile salt solution. ^C^ Bile salt hydrolase (zone size): + 1–2 mm; +++ 3–4 mm; +++++ >5 mm. ^D^ Positive control strain. ^a–f^ Means ± SE with different letters within same column differ significantly at *p* < 0.05.

**Table 2 microorganisms-11-02285-t002:** Antibacterial activity of *Lat. curvatus* strains isolated from *jangajji*.

Strains	Pathogens								
	*S. aureus* ^a^	*S. epidermidis* ^b^	*P. aeruginosa* ^c^	*Pro. acnes* ^a^	*P. putida* ^a^	*S. xylosus* ^a^	*E. coli* ^a^	*B. cereus* ^a^	*B. vallismortis* ^c^
*Lcb. rhamnosus* GG	++	++	++	-	++	+++++	++	+++	++
*Lat. curvatus* JBCC38	+	++	±	-	++	+++	++	+	+
*Lat. curvatus* JBCC47	+	+	±	+	+	++	-	+	+
*Lat. curvatus* JBCC60	±	+	-	+	-	+++	+	+	±
*Lat. curvatus* JBCC65	+	+	+	±	-	++	+	++	+
*Lat. curvatus* JBCC93	+	+++	±	+	-	+++	+	++	++++
*Lat. curvatus* JBCC99	±	++	±	±	±	++	-	+	-

Not clear zone: -, 0 ≤ C ≤ 1: ±, 1 ≤ C ≤ 2: +, 2 ≤ C ≤ 3: ++, 3 ≤ C ≤ 4: +++, 4 ≤ C ≤ 5: ++++, 5 ≤ C: +++++, Control strain: *Lcb. rhamnosus* GG. ^a^: Incubation 10 h, ^b^: incubation 15 h, ^c^: incubation: 20 h.

**Table 3 microorganisms-11-02285-t003:** Antibiotic susceptibility of *Lat. curvatus* strains isolated from jangajji.

Strains	MIC ^a^ (μg/mL)			
	Ampicillin (A)	Chloramphenicol (C)	Vancomycin (V)	Kanamycin (K)
*Lcb. rhamnosus* GG	<2	<2	64	≥1024 ^R^
*Lat. curvatus* JBCC38	<2	<2	≥1024 ^R^	≥1024 ^R^
*Lat. curvatus* JBCC47	<2	2	≥1024 ^R^	512 ^R^
*Lat. curvatus* JBCC60	<2	<2	≥1024 ^R^	512 ^R^
*Lat. curvatus* JBCC65	<2	<2	512 ^R^	32 ^R^
*Lat. curvatus* JBCC93	<2	<2	256 ^R^	256 ^R^
*Lat. curvatus* JBCC99	<2	<2	512 ^R^	64 ^R^

^R^ Resistance according to the EFSA’s breakpoints (EFSA, 2012). ^a^ MIC: minimum inhibitory concentration.

## Data Availability

Not applicable.

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
