# Peer review of "Isolation of Latilactobacillus curvatus with Enhanced Nitric Oxide Synthesis from Korean Traditional Fermented Food and Investigation of Its Probiotic Properties"

_microorganisms, 2023, doi:10.3390/microorganisms11092285_

Round 1

Reviewer 1 Report

In manuscript "Isolation of Latilactobacillus curvatus with Enhanced Nitric  Oxide Synthesis from Korean Traditional Fermented Food and  Investigation of its Probiotic Properties" isolates of L. curvatus with potential probiotic properties were characterized. although this is one of the many studies in which potentially probiotic bacteria are isolated from traditional food.

It is not clearly emphasized which of the mentioned strains has potential as a probiotic bacterium. Check the writing of the abbreviations of the Latin name of the bacteria.

The tables are unreadable and the font should be reduced.

The materials and methods do not list the pathogenic bacteria on which the antimicrobial effect was tested. Are they clinical isolates or standard strains? What medium were they grown on? were antibiotics used as control of the procedure?

Author Response

To Reviewer 1

We thank your thorough and constructive comments. We have addressed each comment in detail below, and the changes made to manuscript are highlighted in the text. We kindly request your reconsideration of our manuscript for publication in Microorganisms.

Q1) It is not clearly emphasized which of the mentioned strains has potential as a probiotic bacterium.

R1) We thank the reviewer for providing valuable comments and suggestions that have helped us improve the quality of our submitted manuscript. As we described in the title and manuscript, we tried to Isolation of LAB with Enhanced Nitric Oxide Synthesis from Jangajji, which is a useful microbial resource for isolating probiotic strains due to its halotolerant and low pH environment. The isolated six strains in this study showed enhanced NO and IL-6 production compared to that of Lcb. rhamnosus GG. Lat. curvatus JBCC38 and JBCC47 induced secretion of higher levels of TNF-α by RAW 264.7 cells. We concluded that Lat. curvatus JBCC93 and JBCC65, hold potential as probiotic candidates due to their ability to produce nitric oxide and stimulate immune responses, as indicated by their effect on IL-6 and TNF-α production in cells.

Q2) Check the writing of the abbreviations of the Latin name of the bacteria.

R2) Thanks. we have re-examined the writing of the abbreviations of the Latin name of the bacteria and changes made to the manuscript are highlighted in the text.

Q3) The tables are unreadable and the font should be reduced.

R3) We have modified the table with reduced font size. Thanks.

Q4) The materials and methods do not list the pathogenic bacteria on which the antimicrobial effect was tested. Are they clinical isolates or standard strains? What medium were they grown on? were antibiotics used as control of the procedure?

R4) Thank you for your question. We used standard pathogens not clinical isolates. Each pathogenic strains were grown in nutrient broth at 37 °C for 24 h. We did not test antibiotics as a positive control for antimicrobial activity, but we used Lcb. rhamnosus GG which currently used broadly at industry to compare the antibacterial activity with our selected strains.

Thanks.

Reviewer 2 Report

REVIEW

Dear authors,

The work proposes the use of strains of Latilactobacillus curvatus (Lat. curvatus) isolated from Jangajji (Korean traditional fermented food) as potential probiotics, with special focus on their immunostimulant capacity through the production of nitric oxide and inflammatory cytokines (IL-6 and TNF-a), which together with the other demonstrated properties, meet most of the criteria to function as potential probiotics.

The reading of the text is easy to understand and the experimental approach is well proposed according to the objectives. However, please consider the following comments to improve the content of your manuscript before publication.

Line 24: write the genus name as it appears throughout the text “Latilactobacillus” instead of “Lactobacillus”.

Line 41: after the genus name they should abbreviate it “Latilatobacillus (Lat) curvatus”.

In Section 1. Introduction, information about Jangajji fermented food is missing.

Lines 74, 175: add a space in “16SrDNA”.

Line 83: is larger than the rest of the text “lipopolysaccharides (LPS:1 g/mL)”. What was the actual amount of LPS added to each well?

Lines 112, 364: write in cursive “Lactobacillus”.

Lines 117, 118, 234, 367: write in cursive “in vitro”.

Line 119: How many Caco-2 cells did they use per well? It does not indicate it.

In the equation correct “Adhesion to Caco-2 = (Absinintial-Absfinal)/Absinintial = D Abs/Absinintial”.

In the equation correct “Surface Hydrophobicity (%)= Absinintial = D Abs/Absinintial”.

Line 144: correctly write drive “mg/ml”.

Line 145: correct symbol position “º”.

Line 169: write correctly and according to the text “Lactolactibacillus”.

Line 179: add a space in “3h”.

Line 182: add a space in “>5mm”. 

Line 185: write in cursive the section“3.2 Evaluation of Nitric Oxide Synthesis Activity of Lat. Curvatus JBCC strains”.

Figure 2: there was no significant difference between the strains?

Line 262: remove the dot symbol in “Propionibacterium. acnes”.

Line 280: write in cursive “L. rhamnosus GG”.

Line 281: write the letters a, b and c in superscript, since in Table 2 the delimitation is not distinguished regarding the names of the pathogens.

Why did you use different incubation times for the tested pathogens?

Write in cursive the scientific names of the microorganisms and abbreviate the name of the journals in the References section.

Please amend the requested comments and submit the revision file.

Author Response

To Reviewer 2

We thank for your thorough and constructive comments. We have addressed each comment in detail below, and the changes made to manuscript are highlighted in the text. We kindly request your reconsideration of our manuscript for publication in Microorganisms.

Q1) Line 24: write the genus name as it appears throughout the text “Latilactobacillus” instead of “Lactobacillus”.

R1) Thanks for pointing out the spelling error. We changed “Latilactobacillus” in line 24 and marked in the manuscript of the revised manuscript.

Q2) Line 41: after the genus name they should abbreviate it “Latilatobacillus (Lat) curvatus”.

R2) Thanks. We changed it.

Q3) In Section 1. Introduction, information about Jangajji fermented food is missing.

R3) Thanks for your comment, it has been added to the Introduction lines 57-59.

Q4) Lines 74, 175: add a space in “16SrDNA”.

R4) Thanks. We added a space “16S rDNA”.

Q5) Line 83: is larger than the rest of the text “lipopolysaccharides (LPS:1 g/mL)”. What was the actual amount of LPS added to each well?

R5) We changed it. We used actually 5 μg to each well

Q6) Lines 112, 364: write in cursive “Lactobacillus”.

R6) Thanks. We changed it.

Q7) Lines 117, 118, 234, 367: write in cursive “in vitro”.

R7) We changed them all through the paper. Thanks

Q8) Line 119: How many Caco-2 cells did they use per well? It does not indicate it.

R8) We inserted the concentration we used initially at this work like this: “We initially seeded Caco-2 cells in each well at 5 × 104 cells/mL.”

Q9) In the equation correct “Adhesion to Caco-2 = (Absinintial-Absfinal)/Absinintial = D Abs/Absinintial”.

In the equation correct “Surface Hydrophobicity (%)= Absinintial = D Abs/Absinintial”.R9) Thanks. We changed them.

Q10) Line 144: correctly write drive “mg/m l”.

R10) Thanks. We changed it.

Q11) Line 145: correct symbol position “º”.

R11) Thanks. We fixed it.

Q12) Line 169: write correctly and according to the text “Lactolactibacillus”.

R12) Thanks for pointing out the spelling error. We changed it.

Q13) Line 179: add a space in “3h”.

R13) Thanks. We added a space“3 h”.

Q14) Line 182: add a space in “>5mm”. 

R14) Thanks. We added a space“> 5mm”.

Q15) Line 185: write in cursive the section“3.2 Evaluation of Nitric Oxide Synthesis Activity of Lat. Curvatus JBCC strains”.

R15) Thanks. We added italics.

Q16) Figure 2: there was no significant difference between the strains?

R16) Sorry, We added significant difference

Q17) Line 262: remove the dot symbol in “Propionibacterium. acnes”.

R17) Thanks. we removed it

Q18) Line 280: write in cursive “L. rhamnosus GG”.

R18) Thanks. We changed it. And the abbreviation form was modified to “Lcb. rhamnosus GG”

Q19) Line 281: write the letters a, b and c in superscript, since in Table 2 the delimitation is not distinguished regarding the names of the pathogens.

Why did you use different incubation times for the tested pathogens?

R19) Thank you for pointing out the error. We have modified Table 2 and added a subscript. We used three different incubation time depend on the pathgens used in this strains due to their growth different.

Q20) Write in cursive the scientific names of the microorganisms and abbreviate the name of the journals in the References section. Please amend the requested comments and submit the revision file.

R20) Thanks for your comments, have modified as requested.

Round 2

Reviewer 1 Report

I have no further comments.